# Paraoxonase-1 as a Regulator of Glucose and Lipid Homeostasis: Impact on the Onset and Progression of Metabolic Disorders

**DOI:** 10.3390/ijms20164049

**Published:** 2019-08-19

**Authors:** Maria João Meneses, Regina Silvestre, Inês Sousa-Lima, Maria Paula Macedo

**Affiliations:** 1CEDOC—Chronic Diseases Research Center, NOVA Medical School/Faculdade de Ciências Médicas, Universidade NOVA de Lisboa, 1150-082 Lisbon, Portugal; 2ProRegeM PhD Programme, NOVA Medical School/Faculdade de Ciências Médicas, Universidade NOVA de Lisboa, 1150-082 Lisbon, Portugal; 3Faculdade de Ciências e Tecnologias, Universidade NOVA de Lisboa, 2829-516 Caparica, Portugal; 4APDP Diabetes Portugal–Education and Research Center (APDP-ERC), 1250-203 Lisbon, Portugal; 5Medical Sciences Department and iBiMED, University of Aveiro, 3810-193 Aveiro, Portugal

**Keywords:** PON1, glucose, insulin resistance, diabetes *mellitus*, NAFLD, obesity

## Abstract

Metabolic disorders are characterized by an overall state of inflammation and oxidative stress, which highlight the importance of a functional antioxidant system and normal activity of some endogenous enzymes, namely paraoxonase-1 (PON1). PON1 is an antioxidant and anti-inflammatory glycoprotein from the paraoxonases family. It is mainly expressed in the liver and secreted to the bloodstream, where it binds to HDL. Although it was first discovered due to its ability to hydrolyze paraoxon, it is now known to have an antiatherogenic role. Recent studies have shown that PON1 plays a protective role in other diseases that are associated with inflammation and oxidative stress, such as Type 1 and Type 2 Diabetes Mellitus and Non-Alcoholic Fatty Liver Disease. The aim of this review is to elucidate the physiological role of PON1, as well as the impact of altered PON1 levels in metabolic disorders.

## 1. Introduction

Metabolic disorders, including Obesity, Type 2 Diabetes *Mellitus* (T2DM), and Non-Alcoholic Fatty Liver Disease (NAFLD), have high prevalence rates worldwide, which makes them primordial concerns for health policy makers [1,2,3]. The last decades have seen a marked increase in overnutrition alongside physical inactivity and sedentarism, which correlates with the growing number of individuals diagnosed with metabolic disorders. When considering that these disorders are characterized by an overall proinflammatory and prooxidant milieu [4,5], the study of endogenous antioxidants and whether they can act as potential biomarkers or therapeutic targets in the management of these disorders is of uttermost importance. The paraoxonase (PON) family is composed by three proteins: PON1, PON2, and PON3. These proteins are differentially expressed and present different functions [6]. PON1, the most studied member of the family, binds to high-density lipoproteins (HDL) and potentiates its antioxidant properties towards low-density lipoproteins (LDL), thus having antiatherogenic effects [7]. This review aims at summarizing the role of PON1 in whole-body lipid and glucose homeostasis, as well as its importance in metabolic disorders, such as atherosclerosis, Diabetes *Mellitus*(DM), and NAFLD.

## 2. Hallmarks of Metabolic Disorders

Cellular homeostasis and an overall balanced metabolic system are crucial requirements for health. Any dysregulation in this balance may lead to the onset and progression of metabolic disorders. These, namely obesity and prediabetes/T2DM are characterized by a proinflammatory and prooxidant state that can contribute to insulin resistance (IR). In fact, all of these conditions will then interfere with several molecular pathways creating a vicious cycle of metabolic impairment.

Oxidative stress is the result of the imbalance in the prooxidant: antioxidant ratio in favor of the former, leading to increased intracellular levels of reactive oxygen species (ROS). These are normally produced in cells, playing important roles in cell signaling. However, when produced over the antioxidant capacity of the cell, ROS lead to the disruption of redox signaling and to molecular damage, which can ultimately result in cell death [8,9]. Thus, oxidative stress plays an important role in inflammatory and metabolic disorders [8,10]. In fact, oxidative stress was already found to suppress insulin secretion from pancreatic β cells and decrease glucose uptake in muscle and fat, which leads to IR [11,12].

Insulin is released from secretory granules in pancreatic β cells in response to increasing plasma levels of glucose and amino acids, in order to normalize them to physiologic levels [13]. The hypoglycemic action of insulin is achieved through the upregulation of glycogenesis in the liver and promotion of glucose uptake in the skeletal muscle and adipose tissue, via glucose transporters (GLUTs) [14,15]. IR refers to a state where insulin-sensitive cells do not appropriately respond to circulating insulin [16,17]. Pancreatic β cells increase insulin release to overcome the reduced efficiency in insulin action in order to maintain normal blood glucose levels; however, this is accompanied by a chronic and subclinical hyperinsulinemia [18]. With time and IR progression, β-cells become unable to fully compensate the decreased insulin sensitivity, leading to T2DM. This affects normal function of different organs, such as the pancreas, liver, bone, and brain [19,20], thus imprinting a transversal feature on metabolic disorders. In the case of liver disease, hepatic IR is a consequence of an impairment on the insulin signaling pathway and glucose uptake, which leads to decreased glycogenesis and hyperglycemia [21,22]. Furthermore, increased blood glucose levels lead to an increased production of ROS, which ultimately results in an overall prooxidant state [23]. In fact, oxidative stress is known as a disease trigger, which plays a significant role in the development of fibrogenesis and inflammation [24].

Inflammation is a protective and complex biological response of the organism to noxious stimuli. It can be considered acute or chronic, with the latter being associated with metabolic, cardiovascular, and neurodegenerative diseases [25]. It is considered to be a hallmark of metabolic disorders, being characterized by the secretion of inflammatory inducers that will lead to the inflammatory response. These inducers may be adipokines and chemokines, as is the case of interleukin-6, tumor necrosis factor-α, and monocyte chemoattractant protein-1 (MCP-1) or even advanced glycation end products and oxidized lipoproteins [26].

IR is associated with metabolic abnormalities, such as hyperglycemia and dyslipidemia [27,28], being a hallmark of T2DM [18]. Dyslipidemia, which is characterized by low HDL cholesterol and high triglycerides levels, is associated with increased free fatty acids (FFAs) in the liver and de novo lipogenesis [29,30]. While considering the effect of IR on overall glucose and lipid homeostasis and oxidative stress, it is crucial to understand the detailed molecular mechanisms that are impacted by it. The paraoxonase family, specifically PON1, may pose as a candidate on the onset and progression of IR-associated inflammation and oxidative stress.

## 3. Paraoxonase Family

The paraoxonase enzyme family is constituted by three members: PON1, PON2, and PON3. Human PONs are encoded by three adjacent but distinct genes that are located on chromosome 7 [31]. In mammals, these genes have 81–95% homology and the resultant proteins share 79–95% of their sequence at the amino acid level [32]. Based on the structural homology and from an evolutionary point of view, PON2 is the oldest member of this family, with PON1 and PON3 probably being a result of gene duplication [31]. Despite that, the family was named, due to early studies where PON1 was found to hydrolyze paraoxon, the toxic metabolite of the insecticide parathion [6,33]. However, PON2 and PON3 do not have this ability to degrade xenobiotics [34]. Besides paraoxonase activity, PON1 has lactonase and ester hydrolase activities, being able to hydrolyze thiolactones, unsaturated aliphatic esters, aromatic carboxylic esters, and carbamates [35]. All of the PON family members degrade lipid peroxides in HDL and LDL [34]. In humans, PON1 and PON3 are mainly expressed in the liver being secreted into the bloodstream, where the enzymes are mainly found bound to HDL [36]. Besides the liver, PON3 is also expressed in the kidney [37]. In contrast, PON2 is expressed in several organs, such as liver, lungs, placenta, testis, and heart, but it is not found in circulation [38]. Within the cell, PON2 is located in the nuclear envelope, endoplasmic reticulum, and in the inner mitochondrial membrane [38]. Here, PON2 is found associated with respiratory complex III being essential for the correct function of the electron transport chain [39]. When considering the marked hepatic expression of PON1 in the liver, it is now accepted that this enzyme has a primordial role as an antioxidant enzyme and promoter of lipid homeostasis.

### 3.1. Paraoxonase-1 (PON1): An Antioxidant Enzyme

PON1 (EC 3.1.8.1), which is the most studied member of the paraoxonase family, is classified as an aryldialkylphosphatase [6]. It was first described in the 1940′s due to its ability to hydrolyze organophosphate compounds [40]. The gene that encodes for PON1 is located on chromosome 7 (7q21.3-q22.1) in humans and in the proximal region of chromosome 6 in mice [41,42]. The coding region of the gene has nine exons and eight introns, which contain many polymorphic CA repeats. Remarkably, it appears that the 5-UTR region of the *PON1* gene does not have a TATA box [31]. *PON1* transcription is highly regulated by specificity protein 1 (Sp1), protein kinase C [43,44], and peroxisome proliferator-activated receptors (PPARs) [45]. Sp1, whose binding site is present in the *PON1* gene, represents a positive regulator of PON1 [46]. It was demonstrated in the hepatic cell lines that an increase of Sp1 enhances PON1 transcription; on the other hand, its absence leads to decreased gene transcription [47]. Moreover, protein kinase C, especially the zeta isoform, increases the binding intensity of Sp1 to *PON1* DNA promoter, thus having a role in its transcription [43,47].

More than 200 single nucleotide polymorphisms (SNPs) were already found on the *PON1* gene, contributing to interindividual differences in PON1 concentration and activity [48]. Of those, there are two common polymorphisms in the coding region of the *PON1* gene with different biological consequences. The leucine to methionine substitution at the position 55 (L55M) affects the PON1 concentration: *PON1L* allele presents more mRNA than *PON1M* allele, leading to an increase in PON1 levels [49,50]. On the other hand, the glutamine to arginine substitution at the position 192 (Q192R) correlates with PON1 activity, as it leads to differences in their catalytic activity towards synthetic substrates. In this case, the *PON1R* allele has higher paraoxonase and arylesterase activity than *PON1Q* allele and the latter shows higher lactonase activity than *PON1R* [51,52]. Moreover, several other polymorphisms have been identified in the non-coding region of the *PON1* gene. One of the most important is the cytosine to thymidine substitution at position 107 (C107T), where the *PON1C* allele leads to higher serum levels of PON1 [53]. Besides the genetic factors, diet, age, lifestyle, pharmaceutical interventions, and epigenetic factors also modulate PON1 activity and serum levels [54,55]. Altogether, these factors contribute to the observed interindividual variability of PON1 [56].

PON1 is a 43 kDa glycoprotein that is composed of 354 amino acids, and its structure is a six-bladed β-propeller, where each blade consists of four β-sheets [57]. Moreover, it has two calcium binding sites that are essential for PON1 activity. One is located at the top of the structure and it is thought to be related with the catalytic activity and the other is in the central section and may contribute to the structural stability of the protein [58,59]. In fact, calcium removal with chelating agents irreversibly destroys PON1′s activity and stability [60].

When secreted into the bloodstream, PON1 is found to be mainly associated with HDL [61], as HDL guarantees shelter for the hydrophobic N-terminal region of PON1 in aqueous environment [62]. This stabilizes PON1 and creates a hydrophobic environment that optimizes its function [63]. Besides facilitating its secretion from the liver, HDL may also be essential for the interaction of PON1 with its substrates [62]. One of the questions that remains to be addressed concerns the specificity with which PON1 associates to lipoproteins. In fact, PON1 does not bind to LDL, which is a major plasmatic lipoprotein. However, PON1 is found within chylomicrons [64] and very low-density lipoproteins (VLDL), the other principal lipoprotein subfraction found in plasma, although to a lesser extent when compared to HDL [65]. PON1-HDL is also commonly associated with two lipoproteins, apolipoprotein A-I (apoA-I) and apolipoprotein J (ApoJ) [66]. The former is a major protein component of HDL, which is known to improve the stability of PON1-HDL interaction [62]. ApoJ, which is also known as clusterin, and its multiple isoforms, are encoded by a gene in chromosome 8 in humans. ApoJ is expressed in several tissues and it has been associated with multiple diseases, including metabolic disorders [67,68]. Additionally, ApoJ acts as a cytoprotective extracellular chaperone for antioxidant enzymes, such as PON1 and it plays a key role in inflammation and immune response [67,69].

Although PON1 was first discovered due to its paraoxonase activity, its substrates are man-made and they have no physiological relevance. Thus, PON1 biological activity can be divided into three categories: lactonase, esterase, and phosphotriesterase, consequently being responsible for the catalysis of different substrates [70]. Besides degrading organophosphates, such as paraoxon, arylesters, cyclic carbonates, glucuronides, and estrogen esters [32,62]; this detoxifier protein also hydrolyses a large diversity of lactones, as well as pharmaceutical agents, namely statins [71]. The latter are used in the treatment of cardiovascular disease (CVD) and exert its effects through the inhibition of cholesterol synthesis in cells and decrease of LDL [72]. As PON1 also degrades oxidized phospholipids and therefore plays a role in the antioxidant system, alterations in circulating PON1 levels and activity have been associated with a variety of diseases that involve oxidative stress. In fact, the interest on this protein has increased due to evidences that decreased PON1 activity is related to several non-communicable diseases, such as cancer, obesity, DM, CVD, neurodegenerative diseases, and NAFLD [73,74].

### 3.2. PON1 and Lipid Metabolism

PON1 was already found to be essential for the maintenance of normal levels of lipid metabolism players. In fact, the absence of PON1 in C57BL/6J mice results in a decrease of glycerol, PUFA/MUFA ratio, and 3-hydroxybutirate [75]. Lower levels of these lipid metabolism intermediates, in comparison with normal conditions, suggest a decrease in lipolysis and decreased fatty acid oxidation, which can be attributable to PON1 impairment. The levels of carnitine were also decreased, which suggests depressed fatty acid oxidation. In fact, carnitine is necessary for the transport of long-chain fatty acids into the mitochondria. Moreover, levels of several lysolipids were reduced by PON1 deficiency, pointing towards reduced membrane remodeling and/or breakdown [75]. There is also a reduction in bile acid metabolism, with a decrease in squalene levels, that is essential for the absorption of lipids from diet and hydrophobic vitamins, such as A, D, E, and K [75]. As lipids affect PON1 levels and activity, the drugs that impact on lipid levels are also expected to impact on PON1. In fact, it has already been observed that statins, one of the most important classes of lipid lowering drugs, might preserve PON1 activity due to its antioxidant properties [76]. Moreover, simvastatin, a drug from the same class, was found to impact on PON1 levels, through sterol regulatory element-binding protein 2 (SREBP2). In fact, SREBP2 is known to bind to the PON1 promoter, which leads to its upregulation. This is apparently made in an interactive manner with Sp1 [77].

PON1 was also found to inhibit macrophage cholesterol biosynthesis [78]. In fact, mouse peritoneal macrophages from PON1 knockout mice had increased cellular cholesterol biosynthesis and PON1 administration led to a direct inhibition of macrophage cholesterol biosynthesis. This mechanism is probably related to PON1 phospholipase-A2-like activity, which results in lysophosphatidylcholine formation and the inhibition of cellular cholesterol biosynthesis [78]. Moreover, PON1 stimulates HDL-mediated cholesterol efflux from macrophages and it attenuates oxidized-LDL uptake by macrophages [79,80] (Figure 1).

Taken together, these studies strongly indicate a role for PON1 in lipid metabolism, which shows its importance for whole-body homeostasis and evidencing its role as a putative target for lipid-lowering drugs.

### 3.3. PON1 and Glucose Homeostasis

PON1 is involved in the regulation of glucose metabolism, as it influences fasting blood glucose levels, glucose tolerance, and insulin sensitivity [81]. It has been demonstrated that PON1 upregulates glucose transporter 4 (GLUT4) expression in the muscle in an insulin receptor-independent manner [81]. In fact, PON1 can regulate the expression and translocation of GLUT4 via the inhibition of p38MAPK activity, leading to reduced insulin receptor substrate-1 serine phosphorylation and to enhanced insulin receptor substrate-1 tyrosine phosphorylation. Moreover, it was demonstrated that the PON1 disulphide groups (SH) and PON1 lactonase activity at catalytic site His115 were key players in the increase in GLUT4 expression [81,82]. Besides regulating glucose uptake, PON1 directly regulates some players of glycolysis, namely 3-phosphoglycerate, phosphoenolpyruvate, and lactate. On the other hand, PON1 negatively affects the levels of fructose-1,6-biphosphate in glycolysis, as well as the levels of components of the pentose phosphate pathway, namely ribulose 5-phosphate, ribose 5-phosphate and xylonate [75]. Accordingly, in the absence of PON1, the activity of glycolysis and the Krebs cycle is decreased, and hence the energy that is obtained in these two pathways is impaired. In contrast, the pentose phosphate pathway becomes more activated, which implies the increase of the NADPH generated [75]. As was the case for the role of PON1 in lipid metabolism, data gathered in the past decade have shown the importance of this enzyme in glucose metabolism and homeostasis, adding to the importance of this enzyme in health and disease.

### 3.4. PON1 and Pancreatic β Cells

β cells are responsible for the production, storage, and release of insulin [83]. These cells function as glucose sensors, crucial for the glucose-stimulated insulin secretion [84]. Under hyperglycemic and or hyperlipidemic conditions, the excessive ROS production results in β cell dysfunction or even apoptosis, which leads to impaired insulin production and secretion [85,86]. The ability of β cells to promote insulin secretion is inversely proportional to cellular oxidation [87]. As PON1 protects against oxidative stress, its antioxidant properties are thought to positively correlate with insulin release by the β cell, under high glucose levels [87]. The administration of PON1, prior to streptozotocin-induced diabetes, led to decreased diabetes onset rates and higher insulin levels [87]. In βTC3 cells, PON1 induces insulin secretion and ameliorates cell survival, under hyperglycemia [87]. Thus, PON1 seems to have a cytoprotective effect on β cells, which improves its viability and significantly increasing insulin secretion, in a dose-dependent manner [87]. Moreover, it was observed that, when PON1 is associated with HDL, the antioxidant properties of this lipoprotein promote an additive effect to insulin secretion [87]. This role seems to be dependent on the availability of PON1 SH groups, as blocking the PON1 SH group, at position Cys284, stops PON1 stimulatory effect on insulin secretion when compared to wildtype PON1 [87,88]. SH groups are expressed in β cell membrane, in insulin granules and in the GLUTs where the SH groups perform structural and functional roles. This shows the importance of PON1 SH groups for stimulation of insulin release from β cells. In contrast, the PON1 catalytic activity and its N-terminus do not influence insulin secretion [82,89]. Additionally, insulin in the β cells increased in the presence of PON1, which indicates that PON1 has a key role in the biosynthesis of insulin [87]. In summary, PON1 seems to promote β cell survival and its normal functional role, which enhances insulin production and secretion.

### 3.5. PON1 and Diet

Diet composition has a significant role in PON1 modulation. A lipid enriched diet is closely associated with inflammation and oxidative stress [90,91]. In fact, the increase of dietary lipids results in an increased production of ROS, lipid hydroperoxides, and inflammatory cytokines, such as TNF-α, interleukin 1, and interleukin 6 [37]. These factors contribute to a decrease in *PON1* expression and consequently decreased hepatic PON1 secretion. This reduction of *PON1* mRNA expression and HDL synthesis are direct consequences of decreased PPARδ, induced by elevated levels of free radicals and lipid peroxidation products in liver [37,92]. Thus, serum HDL-PON1 activity appears to be diminished in high-fat diet, which leads to the impaired reverse cholesterol transport [93].

High-sucrose diet influences glucose metabolism, as sucrose hydrolysis into glucose and fructose increases its plasma levels [94]. On the other hand, as pentose and hexose molecules are converted in triglycerides, High-sucrose diet leads to the increase of circulating VLDL, due to hypertriglyceridemia [95,96]. It has been suggested that VLDL may promote the secretion of antioxidant enzymes by the liver [97]. Several in vivo studies have demonstrated that a high-sucrose diet, as soon as two weeks after the beginning of the diet, induces both hyperlipidemia and oxidative stress, which are known to decrease PON1 activity *per se* [94,98]. However, a study that directly evaluated PON1 activity in rats after three to five weeks of high-sucrose feeding observed a significant increase in PON1 activity [99].

The Mediterranean diet is known as one of the best diets protecting against CVDs. It was already demonstrated that following a Mediterranean diet increases postprandial PON1 activity [100]. These results were similar to those that were obtained after the administration of omega-3 polyunsaturated fatty acids to individuals with familial combined hyperlipidemia [101]. The segmentation of the components of the Mediterranean diet showed that these effects may be attributable to the consumption of extra virgin olive oil [102]. Moreover, some components of the olive oil, not only improve PON1 activity, but also its hepatic mRNA expression and protein levels [103]. Identifying and characterizing dietary components that favor PON1 activity and/or expression might lead to the development of products to enhance the protective role of PON1.

### 3.6. PON1 and Disease

#### 3.6.1. PON1 and Atherosclerosis

Atherosclerosis is a lipid-driven chronic inflammatory disease that is caused by the combination of lipid accumulation and immune cells within the atherosclerotic plaque [104]. A key early step in the development of atherosclerosis is the oxidation of serum LDL. The oxidized products are scavenged by macrophages that may transform into foam cells (Figure 1). These will then become fatty streaks in the endothelium, and then form atheromatous plaques [105]. Studies on animal models have demonstrated that overexpression of PON1 leads to an increased resistance to inflammation and atherosclerosis [106], whereas PON1 knockout mice are more susceptible to lipoprotein oxidation and increased inflammation [107]. Moreover, macrophages of PON1 knockout mice present decreased glutathione concentrations and increased oxidative stress markers and [108], which result in increased oxLDL and foam cell formation [109]. One of the first events of plaque formation is the upregulation of MCP-1 secretion by oxLDL. PON1 inhibits the secretion of MCP-1 in several ways: inhibits its secretion from endothelial cells; and, prevents LDL-derived oxidized phospholipid formation, which is known to stimulate adhesion of monocytes to the endothelial cells and MCP-1 production [110]. Moreover, the formation of lysophosphatidylcholine induced by the HDL-associated PON1 hydrolytic action on macrophage phospholipids enhances HDL binding to macrophages, and consequently enhances cholesterol efflux [111] (Figure 1).

Oxidized lipoproteins, such as LDL and HDL, may be converted into inflammatory signals due to their oxidation by ROS [112]. PON1, isolated or in combination with HDL, protects against LDL oxidation that is promoted by copper ion and free radicals, including ROS [113]. This is mainly due to its ability to hydrolyze peroxide phospholipids in the structure of oxidized LDL and to neutralize the effects of atherogenic lipid peroxides [113,114,115]. The main lipid peroxides that are subjected to hydrolysis by PON1 are triglyceride hydroperoxides, truncated oxidized fatty acids from phospholipid, and cholesteryl ester [7]. Lipid hydroperoxides have a great inflammatory component and cause alterations in carbohydrate and lipid metabolism, particularly in the metabolism of hexoses, glycolysis, Krebs cycle, and phospholipid metabolism decreasing phospholipid synthesis and increasing its degradation [116]. Furthermore, it has been suggested that PON1 is inactivated by oxLDL, due to an interaction of the enzyme free sulfhydryl group (cysteine-284) and oxidized lipids that formed during LDL oxidation, such as oxidized phospholipids, oxidized cholesteryl ester, or lysophosphatidylcholine [114]. Therefore, oxLDL is responsible for inactivating PON1 and, consequently, the arylesterase activity of PON1 decreases, as well as the ability of PON1 to protect against LDL oxidation [114]. Undeniably, metabolic disorders with a prooxidant component lead to the increase of ROS and, hence, result in increased oxidative stress and decreased PON1 antioxidant activity and bioavailability. Moreover, as PON1 inhibits the oxidation of LDL, it prevents the upregulation of proinflammatory cytokines and chemokines, such as monocyte chemoattractant protein 1 produced by cells in the vessel wall [113]. Consequently, PON1 reduces the proatherogenic properties of LDL [117], which thus indicates its role as a protector of cell membrane integrity and cardiovascular complications.

Regarding HDL, PON1 stimulates the antiatherogenic effects of HDL in reverse cholesterol transport, by inducing the mobilization of cholesterol within macrophages and promoting its efflux to HDL (Figure 1) [7]. However, the decrease of PON1 activity and chronic inflammation leads to changes in the HDL proteome, ultimately promoting this lipoprotein’s dysfunction, as well as a decrease of its anti-oxidant and anti-inflammatory capacity [118]. Therefore, cholesterol efflux and reverse cholesterol transport become impaired. As the ability of HDL to modulate oxLDL and to protect it from further oxidation decreases, plasma levels of LDL increase. Consequently, the oxidant status also increases. The increase in circulating oxLDL causes a rise of MCP-1, as well as the activation of the endothelial cell [110,119] promoting atherosclerosis development. These set of factors result in an increased risk to develop CVD and associated complications [120,121].

Altogether, these findings strongly associate the PON1 levels to atherosclerosis onset and development, showing the importance of a fully functioning enzyme to, ultimately, prevent plaque formation.

#### 3.6.2. PON1 and Diabetes

DM is a chronic and multifactorial disease that is characterized by chronic hyperglycemia that is associated with IR, impaired insulin secretion, or both [122]. There are two predominant types of DM: type 1 diabetes *mellitus* (T1DM) and T2DM. Despite the similarity between the chronic complications that may arise from T1DM and T2DM, namely CVD, retinopathy, nephropathy, and atherosclerosis [123], both types are very distinct. T1DM is linked to an autoimmune mediated response targeting pancreatic β cells, which leads to the impaired insulin production [124]. The destruction of these cells is progressive and leads to a profound decrease or even the elimination of insulin production. Therefore, patients with T1DM are dependent on exogenous insulin administration [125]. On the other hand, T2DM is mainly characterized by an increase in IR, due to severe hyperglycemia; however, abnormal β-cell function can also be observed [124]. Contrary to T1DM, the key triggers for T2DM are unhealthy eating habits and sedentarism, although genetic factors may also play a role [126]. Prediabetes, which is the most important predictor of T2DM [127], is characterized by impaired fasting glucose and/or impaired glucose tolerance and it can be associated with hypertension, dyslipidemia, endothelial dysfunction and deficient fibrinolysis [128]. Although all of these mechanisms have been implicated in the pathophysiology of DM, the underlying molecular mechanisms of DM continue to be unveiled [129].

Most studies have found that PON1 activity is decreased in both T1DM and T2DM [130,131,132]. Although this decrease is probably independent of PON1 polymorphisms, PON155L has been associated with diabetic retinopathy [133]. Moreover, in healthy individuals, the L55M polymorphism has been linked to impaired β-cell function, impaired glucose disposal, and increased insulin resistance [134,135]. The mechanism by which PON1 is reduced in DM remains poorly understood, but it might be linked to blood glucose levels. Under hyperglycemic conditions, both PON1 expression and activity are affected. High concentrations of hexose molecules are accompanied by IR and increased lipid accumulation, particularly of diacylglycerol [136]. This causes the activation of numerous protein kinase C isoforms [137], which are responsible for the phosphorylation of specific protein 1 (SP1). As SP1 is a positive regulator of *PON1*, its phosphorylation will enhance *PON1* transcription [47]. In metabolic disorders, the presence of several factors that are linked to abnormally high glucose levels, such as hyperinsulinemia, dyslipidemia, and oxidative stress, lead to a decrease of PON1 activity, especially due to glycation of HDL-PON1 [138]. In fact, the mRNA expression and synthesis of serum PON1 are both increased in order to compensate this decrease in PON1 activity [139]. It has also been demonstrated that the levels of PON1 are significantly decreased in IR [140]. Therefore, in both types of DM, serum PON1 activity and concentration are significantly decreased, since hyperglycemia, oxidative stress, and IR coexist [141].

Dyslipidemia is closely associated with diabetes [142], since the latter promotes qualitative, quantitative, and kinetic alterations in circulating lipids [143,144]. Thus, the lipid profile is changed in T2DM, with triglycerides and LDL-cholesterol plasma levels being significantly higher when compared to normal healthy conditions [145]. On the other hand, HDL levels and antioxidant activity appear to be decreased in T2DM. HDL cholesterol affects glucose metabolism, through the alteration of glucose uptake in skeletal muscle and the promotion of insulin release by β cells [146]. It has been shown that the intravenous HDL administration in patients with T2DM led to increased plasma insulin and a consequent decrease glucose plasma levels [146,147], showing that HDL is an important player in glucose homeostasis.

Glycation or non-enzymatic glycosylation of PON1 is another consequence of hyperglycemia that is associated with diabetes mellitus [148]. This process leads to the inhibition of the paraoxonase activity of PON1. Glycation is also responsible for HDL dysfunction and catabolism [149], so glycated HDL is unable to metabolize membrane lipid peroxides and maintain the cholesterol efflux [138]. It was found that HDL isolated from individuals with lower PON1 activity are more susceptible to lipid peroxidation in hyperglycemic conditions than HDL from subjects with higher PON1 activity [150]. It has been demonstrated that glycated PON1 also results in endothelial dysfunction in T2DM through endoplasmic reticulum stress, which is widely associated with other pathologies, such as atherosclerosis, obesity, NAFLD, neurodegenerative disorders, and cancer [151]. This process is triggered by hyperglycemia and oxidative stress [148]. Therefore, in metabolic disorders, the activity of PON1 and anti-inflammatory and anti-atherosclerotic properties of serum HDL-PON1 complex are significantly decreased due to inactive oxidation and glycation of both free PON1 and HDL-PON1.

Hence, the enzyme PON1 has recently been proposed as an intervenient in the pathophysiology of DM, with its normal levels being crucial for whole-body homeostasis.

#### 3.6.3. PON1 and Obesity

Obesity is defined by the World Health Organization (WHO) as abnormal or excessive fat accumulation. There are several factors that may be implicated in the etiology of obesity, namely genetics, environment, and socioeconomic and psychological factors [152]. The body mass index (BMI) is the most used method to classify obesity. An individual is considered obese if the BMI is higher than 30 kg/m^2^ or overweight if the BMI is between 25 and 29 kg/m^2^ [153]. However, and as body fat distribution is a risk factor for the development of comorbidities, other measures, such as waist circumference, might be used in the clinical practice, to refine the diagnosis that was obtained through BMI [154]. In fact, abdominal fat is associated with increased metabolic risk factors and increased risk of cardiometabolic disease [154,155]. Undeniably, obesity causes numerous abnormalities in lipid metabolism, including increased triglycerides, lower levels of HDL, and alterations in lipoprotein structure and concentrations, namely due to the dysregulated production of adipokines [4,10,156]. Adipose tissue also secretes non-esterified fatty acids to the bloodstream, and their flux to the liver may affect liver metabolism, increasing hepatic IR and glucose production [157]. Moreover, proinflammatory cytokines are also secreted, resulting in low-grade inflammation and vascular dysfunction, and thus promote IR [158,159]. Besides the increased risk of T2DM, obesity is linked to other health problems, including NAFLD, atherosclerosis, degenerative disorders, and several types of cancer [160].

PON1 activity was reported to be significantly lower in obese individuals when compared to the controls [161,162]. In fact, this decrease was accompanied by an increase in the levels of lipid hydroperoxides in HDL and LDL from obese subjects, which suggests that obese individuals are more susceptible to oxidative damage [161]. Moreover, BMI was already found to be an independent predictor of PON1 activity [163]. This decrease in PON1 activity in obese subjects is independent of age, as it was also found in young subjects [164]. Moreover, PON1 arylesterase activity was negatively correlated with leptin concentrations, but positively correlated with adipokine levels. As was the case for IR the study of the molecular mechanisms promoting fat accumulation is also fundamental. Again, PON1, with its antiatherogenic properties, might be an ideal candidate in the pathophysiology of obesity.

#### 3.6.4. PON1 and Non-Alcoholic Fatty Liver Disease

The liver has a crucial role in whole-body homeostasis, namely in the regulation of glucose and lipid metabolism. However, in pathophysiological states, these metabolic pathways may be impaired, which leads to hepatic fat accumulation (i.e., steatosis), a hallmark of fatty liver disease. NAFLD is characterized by excessive fat accumulation in the liver with more than 5% of hepatocytes containing visible intracellular triglycerides or steatosis and affecting at least 5% of the liver total volume, in patients without other causes of steatosis (excessive alcohol consumption, hepatitis, glucocorticoids, hypothyroidism, tamoxifen, etc.; Figure 2) [165,166].

NAFLD and T2DM are highly related, as fatty liver causes IR and the impairment of insulin secretion and glucose uptake [167,168]. Indeed, in association with other metabolic dysfunctions, such as T2DM, overweight, and hypertension, NAFLD may progress to a severe and advanced form of liver disease, named non-alcoholic steatohepatitis (NASH) [169,170]. In fact, inflammation, insulin dysregulation, and altered lipid homeostasis can induce lipotoxicity in fatty hepatocytes. The latter release signals that are necessary for the repair of liver damage. Consequently, the repair-related cells, such as immune cells, accumulate and begin the wound-healing response that includes inflammation, fibrogenesis, and hepatic accumulation of immature liver epithelial cells [171,172,173]. NASH is the sum of injury and repair responses that are caused by lipotoxicity [171]. It is characterized by the presence of macrovesicular steatosis, inflammation, and ballooning. Ballooned hepatocytes, which present a cellular degeneration characterized by cellular swelling, reticulated cytoplasm, central nucleus, and disorganized cellular polarity, are considered a hallmark of NASH [174]. NASH is one of the causes of CVD, and may progress to cirrhosis, and end-stage liver disease [175]. In fact, NAFLD progression to NASH and cirrhosis is the major cause of hepatocellular carcinoma, the sixth most common cancer type in the world (Figure 2) [169,176,177,178,179].

As PON1 has a protective effect against oxidative stress and inflammation, it is expected to have a role in NAFLD. In PON1 knockout mice, a high-fat/high-cholesterol diet caused increased oxidative stress and metabolic alterations that lead to the development steatosis [75]. Moreover, these mice presented decreased glycolysis and Krebs cycle, as well as the urea cycle. On the contrary, triglyceride and phospholipid synthesis were significantly increased [75]. A more recent study has demonstrated that serum PON1 activity is paradoxically maintained in patients with fatty liver index (FLI), above 60, despite low HDL [180]. Moreover, a study using paraoxon as substrate demonstrated that PON1 activity was decreased in patients with hepatic steatosis, as compared with healthy subjects [181].

#### 3.6.5. PON1 and Cancer

The growth of a tumor relies on several factors, with an interplay of inflammation and oxidative stress. A proinflammatory and prooxidative environment contributes to cellular damage and it may lead to carcinogenesis. During carcinogenesis, cancer cells have an intricate machinery to regulate lipoprotein uptake, in order to maintain cell metabolism and homeostasis [182]. In the last years, there is a growing interest on the role of PON1 in cancer. Lower activity of PON1 was already associated with several types of cancer, namely lung [183], pancreatic [184], gastric [185], colorectal [186], and ovarian [187] cancer. However, the majority of the studies have a small number of individuals and corresponding healthy controls. Furthermore, the existence of polymorphisms has been mostly disregarded, which may alter the significance of the results. One meta-analysis about PON1 genetic polymorphisms and breast cancer susceptibility has concluded that the presence of a R allele on the polymorphism Q192R was associated with a decreased risk of breast cancer. On the contrary, PON1–55LM and PON1–55MM genotypes were both associated with increased risk [188]. In fact, individuals with MM and QQ genotypes present lower levels of PON1, which may decrease the ability to detoxify inflammatory oxidants, as well as dietary carcinogens, leading to an increased risk of breast cancer development [188]. This once more proves the importance of analyzing PON1 genotype and not only PON1 activity.

#### 3.6.6. PON1 and Alzheimer’s Disease

Sporadic Alzheimer’s disease (AD) is the most common form of dementia in the elderly [189]. It is a neurodegenerative disease that is caused by accumulation of insoluble aggregates of hyperphosphorylated tau protein, in the form of neurofibrillary tangles and neurotoxic amyloid-beta precursor protein peptides, among others [189]. These will then result in the most known symptoms of the disease, cognitive function decline, and memory loss. The perception that AD may be considered a metabolic disorder arise from studies that showed that early cognitive dysfunction could be preceded by deficits in glucose consumption in the brain [190,191]. PON1 has been extensively studied in relation to AD, due to its anti-inflammatory and antioxidant properties. In fact, a study using double knockout (DK) mice for PON1 and ApoE, the major cholesterol carrier in the brain, has demonstrated that aged animals have an increased expression of AD markers in the brain. These 14 months DK mice presented elevated serum levels of S100 calcium-binding protein B, increased expression of genes that are involved in formation of Aβ plaques (β-site amyloid precursor protein cleaving enzyme 1, presenilin 1, and presenilin 2), and proinflammatory genes (IL-1β, IL-6, MCP-1) [192]. Moreover, studies have examined the contribution of PON1 genotypes to AD occurrence. It was already demonstrated an association between AD and variants in the PON gene cluster in Caucasians and African Americans [193]. Specifically, there was significant evidence of an association with AD for the SNP-C161T in both ethnic groups [193], with the T-allele being the one with deleterious effects. However, a study analyzing the same SNP in an Italian case-control population found no association with AD [194]. Another study observed that a heterozygous methionine allele of the L55M polymorphism increases the risk of AD in men. Interestingly, homozygous individuals for both QQ and MM genotype present an increased survival and later age of onset of AD [195]. As was the case for the previously referred diseases, PON1 also has a role in AD progression. By being a key component in lipid metabolism this enzyme is now known to impact on a multitude of pathophysiological processes.

## 4. Conclusions

Metabolic disorders are considered to be one of the most worrying public health issues worldwide, posing a number of socio-economical limitations and representing a burden for national health systems [196]. Oxidative stress and inflammation, which are hallmarks of metabolic disorders, have a profound impact on PON1 levels and activity. The vast majority of circulating PON1 is associated with HDL, contributing to its antiatherogenic properties. Moreover, PON1 levels and activity are significantly impaired in individuals with cardiovascular and liver diseases, as well as in DM and obesity. Taking this in consideration, PON1 might pose as a biomarker for the detection of these diseases. However, the enormous interindividual variability of PON1 makes its use as a biomarker challenging. Furthermore, we also must take in account that different polymorphisms lead increased susceptibility to different disorders. For now, PON1 activity can be increased through lifestyle modifications, as is the case of a Mediterranean diet. In summary, more studies are needed to decipher the mechanisms and functions of PON1, in health and disease. These will also help in discovering new regulators of PON1 levels and activity, in order to avoid metabolic disorders progression into more serious pathophysiological states.

## Figures and Tables

**Figure 1 ijms-20-04049-f001:**
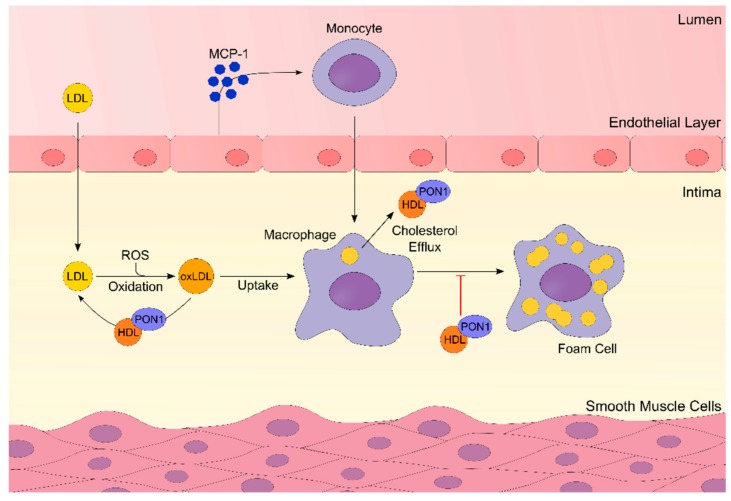
The role of Paraoxonase-1 (PON1) in the pathophysiology of atherosclerosis. Inflammation and oxidative stress lead to the oxidation of low-density lipoproteins (LDL). The release of monocyte chemoattractant protein-1 (MCP-1) from endothelial cells will lead to the recruitment of monocytes. These will differentiate into macrophages, that will internalize the oxidized LDL (oxLDL), becoming foam cells. PON1 hydrolyzes the oxLDL, reverting it to LDL, and promotes cholesterol efflux from macrophages, inhibiting the progression of atherosclerosis. ⊣—inhibition.

**Figure 2 ijms-20-04049-f002:**
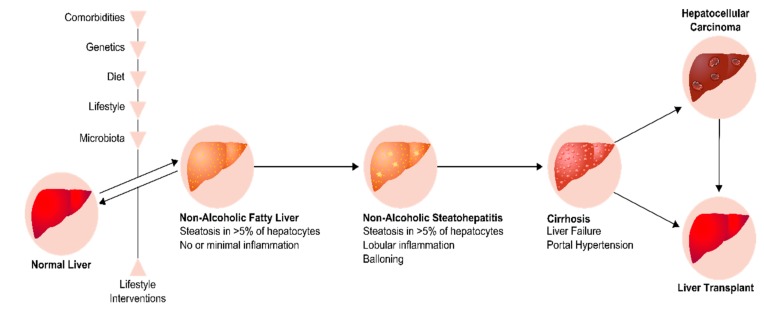
Progression of Non-Alcoholic Fatty Liver Disease. The liver begins to accumulate fat due to erratic lifestyle habits, genetics, microbiota and comorbidities leading to Non-Alcoholic fatty liver. It can then progress to Non-Alcoholic Steatohepatitis (NASH) or be reverted through lifestyle interventions. NASH is characterized by lobular inflammation and ballooning. In some cases, it can progress to cirrhosis. This is a severe stage than can lead to the development of hepatocellular carcinoma (HCC), ending up in the need of liver transplant.

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
