# Peer review of "Paraoxonase-1 as a Regulator of Glucose and Lipid Homeostasis: Impact on the Onset and Progression of Metabolic Disorders"

_ijms, 2019, doi:10.3390/ijms20164049_

Round 1

Reviewer 1 Report

This is an informative review on the pleiotropic biological functions of paraoxonase-1 (PON) and its perturbations in context of metabolic and neurodegenerative diseases. Similar reviews have been published by others. The first portion of the manuscript is written very well, but there are some shortcomings in grammar and syntax in the later parts. Editing of the manuscript by a native speaker may be useful here. Overall, the manuscript will be of interest to the journal’s readership.

Page 4, Line 171: ‘intervenient’ should probably read ‘intermediate’

Page 6, Line 232: correct ‘N-terminal’ to ‘N-terminus‘

Page 7, Line 278: it appears some words are missing in the sentence beginning with ‘In one hand, inhibits…

Page 7, Line 286: typo ‘cooper’, should read copper

Author Response

Lisboa, August 14, 2019

Ref.: Resubmission of the manuscript by Meneses MJ et al. “PON1 as a regulator of glucose and lipid homeostasis: impact on the onset and progression of metabolic disorders” (Manuscript ID ijms-572724)

Response to Reviewers:

We thank the reviewers for the very pertinent observations and suggestions. We have altered the manuscript accordingly and all the alterations are addressed bellow for each one of the reviewers. Alterations were signalized in the manuscript.

All the previous suggestions were considered and are signalized in the manuscript. We thank the reviewer for the suggestions.

Sincerely,

M Paula Macedo

Reviewer 2 Report

In this review, Meneses and colleagues collect the knowledge regarding the role of PON1 in metabolic disorders. I read your manuscript with great interest. All sections are well defined and this is potentially interesting study, and I have some minor suggestions to make the manuscript more comprehensive.

Minor comments:

    Many studies have pointed out that PON1 is associated with oncometablism, and the author should discuss this topic in the disease section.

    The overall flow of the article reads like a number of statements rather than a story. I would recommend working on transitioning between sections better.

Author Response

Lisbon, August 14, 2019

Ref.: Resubmission of the manuscript by Meneses MJ et al. “PON1 as a regulator of glucose and lipid homeostasis: impact on the onset and progression of metabolic disorders” (Manuscript ID ijms-572724)

Response to Reviewers:

We thank the reviewers for the very pertinent observations and suggestions. We have altered the manuscript accordingly and all the alterations are addressed bellow for each one of the reviewers. Alterations were signalized in the manuscript.

Reviewer #2:

Many studies have pointed out that PON1 is associated with onco-metablism, and the author should discuss this topic in the disease section.

We thank the reviewer for the suggestion. We followed this suggestion and a section about “PON1 and cancer” was included in the manuscript.

The overall flow of the article reads like a number of statements rather than a story. I would recommend working on transitioning between sections better.

The authors thank the reviewer for the observation. We improved the transition between sections and subsections.
